# JADE: Bridging the Strategic-Operational Gap in Dynamic Agentic RAG

**Yiqun Chen** [* 1]  **Erhan Zhang** [* 1]  **Tianyi Hu** [* 2]  **Shijie Wang** [* 3]  **Zixuan Yang** [1]  **Meizhi Zhong** [4]  **Xiaochi Wei** [4]  **Yan Gao** [4]  **Yi Wu** [4]  **Yao Hu** [4]  **Jiaxin Mao** [† 1]

## Abstract

The evolution of Retrieval-Augmented Generation (RAG) has shifted from static retrieval pipelines to dynamic, agentic workflows where a central planner orchestrates multi-turn reasoning. However, existing paradigms face a critical dichotomy: they either jointly optimize modules within rigid, fixed-graph architectures, or enable dynamic planning while treating executors as frozen, black-box tools. We identify that this *decoupled optimization* creates a "strategic-operational mismatch," where sophisticated planning strategies fail to materialize due to unadapted local executors, often causing negative gains despite increased system complexity. In this paper, we propose **JADE** (**J**oint **A**gentic **D**ynamic **E**xecution), a unified framework for joint optimization of planning and execution within dynamic, multi-turn workflows. By modeling the system as a cooperative multi-agent team with a shared backbone, JADE enables end-to-end learning driven by outcome-based rewards. This approach facilitates *co-adaptation*: the planner learns to operate within executor capability boundaries, while executors evolve to align with strategic intent. Empirical results demonstrate that JADE transforms disjoint modules into a synergistic system, yielding strong performance improvements via joint optimization and enabling a flexible balance between efficiency and effectiveness through dynamic workflow orchestration.

## 1. Introduction

The integration of Large Language Models (LLMs) with external knowledge bases has catalyzed a shift from sim-

---

[*]Equal contribution  [1]Renmin University of China [2]Institute of Automation, Chinese Academy of Sciences [3]Shanghai AI Laboratory [4]Xiaohongshu Inc.. Correspondence to: Jiaxin Mao <maojiaxin@gmail.com>.

*Proceedings of the 43rd International Conference on Machine Learning*, Seoul, South Korea. PMLR 306, 2026. Copyright 2026 by the author(s).

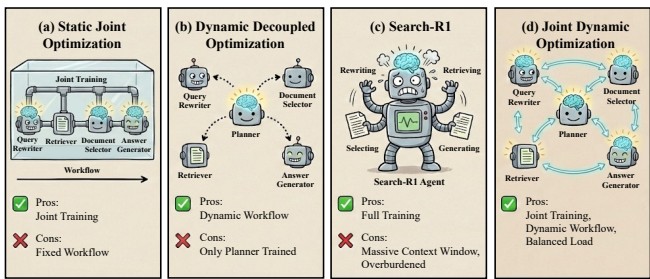

*Figure 1.* Different Paradigms of Agentic RAG

ple Retrieval-Augmented Generation (RAG) to autonomous *Agentic RAG* (Shi et al., 2025) systems. These systems aim to solve knowledge-intensive tasks not merely by retrieving documents, but by actively planning and executing multi-step reasoning trajectories. Despite rapid progress, current approaches struggle to balance architectural flexibility with optimization stability. As illustrated in Figure 1, existing paradigms comprise three distinct classes, each facing critical limitations that necessitate a new modeling perspective.

The first paradigm, **Static Joint Optimization** (Figure 1(a)), characterizes early modular RAG systems (Chen et al., 2025a). These architectures define a fixed computational graph—typically a linear sequence of Query Rewriting, Document Selection, and Answer Generation. While these modules are optimized jointly to maximize system performance, the rigid topology restricts the agent to a "one-size-fits-all" workflow. Consequently, such systems lack the adaptivity required to decompose complex, multi-hop queries that demand variable reasoning paths.

To address this rigidity, the field advanced toward **Dynamic Decoupled Optimization** (Figure 1(b)). These methods (Chen et al., 2025b; Jiang et al., 2025a; Mei et al., 2025) introduce a centralized *Planner* to dynamically orchestrate workflows. However, these systems adopt a decoupled training strategy: the Planner is optimized to generate high-level plans, while the Executors (e.g., retrievers, readers) are treated as frozen black-box tools. We identify this separation as a source of *strategic-operational mismatch*. As depicted in Figure 1(b), the Planner may devise sophisticated strategies that disjoint Executors are ill-equipped to realize, leading to execution failures and suboptimal global performance despite theoretically sound planning.

Most recently, unstructured reasoning models like **Search-R1** (Figure 1(c)) have emerged, attempting to fuse planning, search, and answer generation into a single end-to-end stream (Jin et al., 2025; Song et al., 2025a;b; Zheng et al., 2025). While this removes the constraints of modular architectures, it introduces training instability (Deng et al., 2025; Chen et al., 2026). The model must simultaneously learn to reason, query noisy search engines, and filter information within a massive context window. Lacking structural priors, the optimization landscape becomes perilous; the agent often struggles to converge, burdened by the cognitive load of managing the entire process of the agentic search implicitly.

In this paper, we propose **JADE** (**J**oint **A**gentic **D**ynamic **E**xecution), a unified framework designed to harmonize flexibility and stability. As shown in Figure 1(d), JADE retains the modular clarity of planner-executors architectures but fundamentally redefines their interaction as a *cooperative multi-agent collaboration*. Unlike decoupled approaches (Figure 1(b)), JADE optimizes the Planner and Executors simultaneously via shared parameters within a single LLM backbone driven by sparse global rewards. This *Joint Dynamic Optimization* fosters co-adaptation: the Planner learns to orchestrate workflows that respect the functional boundaries of the Executors, while the Executors evolve to align with the Planner's high-level strategic intent. By transforming disjoint modules into a synergistic team, JADE combines the adaptive power of dynamic planning with the convergence stability of joint optimization.

Our contributions are summarized as follows:

- We introduce **JADE**[1], a novel framework that formulates multi-turn information seeking as a cooperative multi-agent game. By enabling end-to-end gradient flow across the Planner and Executors, JADE bridges the gap between high-level reasoning and low-level execution.
- Empirical evaluations demonstrate that JADE establishes a new SOTA on seven benchmarks, effectively balancing computational cost with task effectiveness. Notably, our jointly optimized 7B model outperforms GPT-4o-based decoupled systems, demonstrating that collaborative synergy is more critical than raw model scale for complex reasoning.

**Conflict of Interest Disclosure.** The authors declare no financial conflicts of interest related to this work.

## 2. Related Work

**From Static RAG to Dynamic RAG.** Early Retrieval-Augmented Generation systems typically relied on *static retrieval pipelines*, where retrieval and generation occur in

---

[1] The source code for JADE is available at https://github.com/chenyiqun/JADE.

a fixed, pre-defined sequence. Representative works such as RALM (Xia et al., 2025), LongRAG (Zhao et al., 2024), IN-STRUCTRAG (Wei et al., 2024), RRR (Ma et al., 2023), and BGM (Ke et al., 2024) focus on enhancing specific modules within this static paradigm but lack the flexibility to adjust the retrieval strategy based on query complexity. To address multi-hop reasoning, *iterative frameworks* introduced interleaved retrieval and generation steps. Approaches like IRCoT (Trivedi et al., 2023), Self-RAG (Asai et al., 2023), Adaptive RAG (Jeong et al., 2024), and ReSP (Jiang et al., 2025b) allow for dynamic loop control or self-reflection. However, these methods primarily rely on heuristic control flows or supervised fine-tuning without employing end-to-end reinforcement learning, limiting their ability to discover globally optimal strategies in complex environments.

**Optimization Paradigms for Agentic Search Systems.** Recent advancements in Agentic Search have adopted Reinforcement Learning (RL) to enhance decision-making, though different paradigms exhibit distinct trade-offs. One dominant approach focuses on *decoupled planner optimization*. Frameworks such as MAO-ARAG (Chen et al., 2025b), S3 (Jiang et al., 2025a), and AI-SEARCHPLANNER (Mei et al., 2025) utilize RL to train a specialized Planner to orchestrate dynamic workflows. By tailoring the reasoning path to the query complexity, these methods allow for an adaptive trade-off between effectiveness and efficiency. However, they treat executors as frozen black boxes, leading to strategic-operational misalignment. Conversely, MMOA-RAG (Chen et al., 2025a) achieves *joint optimization* by simultaneously training the Query Rewriter, Document Selector, and Generator using RL. While synergistic, MMOA-RAG is constrained to a fixed, single-turn workflow, restricting its applicability to long-horizon tasks. A third paradigm, exemplified by Search-R1 (Jin et al., 2025) and similar methods (Zheng et al., 2025; Song et al., 2025a;b), employs *monolithic* RL to generate entire reasoning chains and search actions end-to-end. While this reasoning-enhanced, iterative paradigm is highly flexible, the confluence of long-horizon generation, sparse reward signals, and noise introduced by external search engines significantly complicates the optimization landscape of RL training. Consequently, these models often suffer from severe training instability and convergence to suboptimal solutions (Deng et al., 2025; Chen et al., 2026). **JADE** synthesizes these approaches by applying the joint optimization of MMOA-RAG to the dynamic, multi-turn workflows of MAO-ARAG, offering a structured, modular alternative to the monolithic nature of Search-R1.

**Multi-Agent Reinforcement Learning (MARL).** Our framework draws theoretical grounding from cooperative Multi-Agent Reinforcement Learning. Classic works in this domain (Rashid et al., 2020; Lowe et al., 2017; Yu et al., 2022; Chen et al., 2022) typically utilize parameter shar-

ing and model the environment as a fully cooperative task, where all agents are optimized via a shared global reward to maximize total team utility. Similarly, we formulate the internal modules of an LLM as a multi-agent team. Recognizing that Agentic RAG is inherently a partially observable scenario (POMDP) (Kaelbling et al., 1998) where agents only perceive limited retrieval contexts, JADE employs parameter sharing and a unified global reward to incentivize distinct functional roles (Planner and Executors) to co-adapt. This effectively solves the temporal credit assignment problem in long-horizon reasoning tasks, ensuring that local execution aligns with global strategic objectives.

## 3. Methodology

In this work, we propose **JADE** (**J**oint **A**gentic **D**ynamic **E**xecution), a framework that unifies strategic planning and operational execution into a single, end-to-end learnable policy. Unlike prior decoupled approaches where the planner is optimized against fixed, black-box executors, JADE employs homogeneous parameter sharing to facilitate co-adaptation between high-level workflow planner and low-level executors.

### 3.1. Problem Formulation: Shared-Parameter MSMDP

We formulate the dynamic retrieval-augmented generation process as a Multi-Agent Semi-Markov Decision Process (MSMDP) (Ghavamzadeh et al., 2006) with partial observability. The system is defined by the tuple $\langle \mathcal{S}, \Omega, \mathcal{A}, \mathcal{P}, \mathcal{R}, \gamma, \mathcal{T} \rangle$.

**Global State Space** ($\mathcal{S}$). The global state acts as the fully observable environment or "blackboard" that records the evolving history of the collaborative reasoning process. We formalize the global state $s_t \in \mathcal{S}$ at the beginning of **round** $t$ as a structured tuple:

$$s_t = \{Q_{origin}, \mathcal{T}_t\} \tag{1}$$

where $Q_{origin}$ is the initial user query, and $\mathcal{T}_t$ is the **dynamic execution trace**. The trace maintains an ordered sequence of task nodes, which expands as the Planner decomposes the problem. Each node $n_m \in \mathcal{T}_t$ (indexed by $m$) is defined as a tuple:

$$n_m = \langle q_m, a_m \rangle \tag{2}$$

Here, $q_m$ denotes the specific sub-query derived from decomposition, and $a_m$ represents the answer to that sub-query (initialized as $\emptyset$). As agentic search progresses within round $t$, new nodes may be appended, and empty answer slots are populated sequentially.

**Observation Space** ($\Omega$). The inference process within round $t$ involves a sequence of steps indexed by $k =$ $0, \ldots, K_t$. To ensure computational efficiency while maintaining context awareness, agents operate on a **role-specific observation** $o_{t,k} \in \Omega$.

We define Context$_{t,k}$ as the **intra-round working memory**, which accumulates the intermediate outputs generated by preceding agents (steps $0$ to $k-1$) within the current round (e.g., the workflow $w$, or retrieved documents).

The observation function $\mathcal{O}$ constructs the input by combining the **current target sub-query** $q_{target}$ with the local context, augmented by relevant information selectively retrieved from the global state $s_t$ based on the agent's active role $\rho_{t,k}$:

$$o_{t,k} = \mathcal{O}(\underbrace{q_{target}}_{\text{Current Goal}} \cup \underbrace{\text{Context}_{t,k}}_{\text{Local Updates}} \cup \underbrace{\text{Select}(s_t, \rho_{t,k})}_{\text{Global History}}, \rho_{t,k}) \tag{3}$$

Here, $s_t$ serves as the global memory bank. The function Select$(\cdot)$ filters $s_t$ to provide necessary historical context (e.g., previously resolved sub-answers) without overwhelming the agent with irrelevant trace details. For instance, a *Query Rewriter* may access resolved answers in $s_t$ to resolve coreferences, while a *Document Selector* focuses primarily on the immediate retrieval documents.

**Hierarchical Action Space** ($\mathcal{A}$). The action space is a union of heterogeneous sub-spaces: $\mathcal{A} = \mathcal{A}_{\text{plan}} \cup \mathcal{A}_{\text{exe}}$.

The *Planner Action Space* $\mathcal{A}_{\text{plan}}$ is discrete but combinatorial. At the start of a round ($k = 0$), the Planner generates a structured execution plan $w \in \mathcal{A}_{\text{plan}}$. Each plan $w$ involves selecting a subset of executors $\mathcal{E} \subseteq \mathcal{R}_{exec}$ and orchestrating their directed topology (e.g., sequential chains or parallel groups) to form an executable workflow graph. This allows the Planner to deploy complex maneuvers, such as " Decompose $\rightarrow$ Parallel Retrieval $\rightarrow$ Summarize", in a single strategic decision step.

The *Executor Action Space* $\mathcal{A}_{\text{exe}}$ is semantic. In subsequent steps ($k > 0$), the activated agents defined in workflow $w$ generate specific operational outputs to populate the respective nodes.

**Unified Policy and Shared Objectives.** To enable effective coordination, we unify the agent space using a single LLM backbone parameterized by $\theta$.[2] The policy conditions on the step-specific observation rather than the full state:

$$a_{t,k} \sim \pi_\theta(\cdot | o_{t,k}, p_{\rho_{t,k}}) \tag{4}$$

where $p_{\rho_{t,k}}$ is the specific system prompt corresponding to role $\rho_{t,k}$. The optimization objective is to maximize the joint expected return based on a global reward $R$ shared by

---

[2]We provide a detailed rationale for this parameter-sharing strategy, covering its theoretical foundations in MARL and deployment efficiency, in Appendix A.

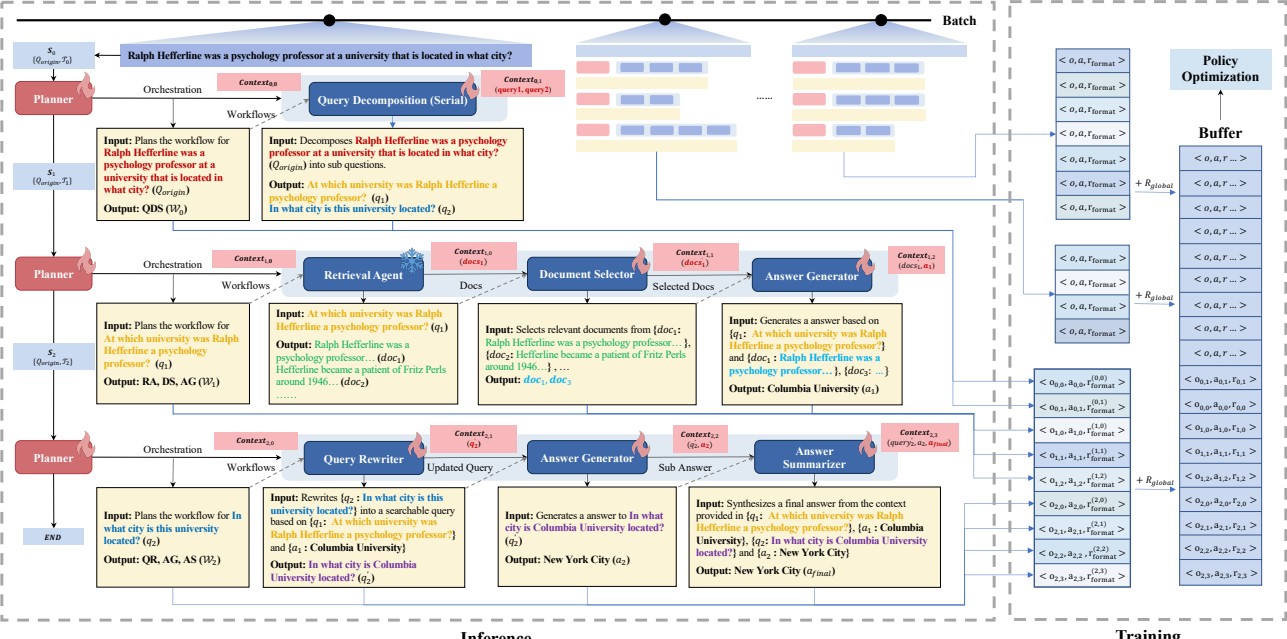

*Figure 2.* The overall framework of JADE. The system operates in an iterative loop of planning and execution. (1) During **Inference** (Left), the process is recursive: for the current unsolved node (i.e., a specific sub-query, see Eq.2) in the global state $s_t$, the **Planner** is invoked to orchestrate a dedicated dynamic workflow. This workflow is **executed** by specialized **Executors** (e.g., Query Decomposition, Retrieval Agent) to update $s_t$ for the next round. (2) During **Training** (Right), to achieve joint optimization, every agent involved in the multi-turn trajectory generates transition triplets $\langle o_{t,k}, a_{t,k}, r_{t,k} \rangle$. These transitions are aggregated into a unified **Experience Buffer**, which is then used to update the parameter-sharing policy model, aligning strategic planning with operational execution.

all agents in the trajectory $\tau$:

$$\mathcal{J}(\theta) = \mathbb{E}_{\tau \sim \pi_\theta} \left[ \sum_{t=0}^{T} \sum_{k=0}^{K_t} \gamma^{(t,k)} R(s_{t,k}, a_{t,k}) \right] \quad (5)$$

Here, the inner sum represents the steps within a dynamic workflow, and the outer aggregates across reasoning rounds.

### 3.2. Agentic Roles and Workflow

Building upon the MSMDP formulation, JADE implements its agentic space not as static, disparate tools, but as distinct *trainable personas* derived from the shared backbone $\pi_\theta$. Each role $\rho$ is instantiated by conditioning the policy on a specific system instruction. The specific definitions are formalized in Table 1[3], and the detailed system prompts for each role are provided in Appendix E.

The inference process[4] is formalized in Algorithm 1 (see Appendix B for detailed pseudocode) and visually illustrated in Figure 2 (Left). Initialized with the raw query and a root trace node, the system enters an iterative loop until the trace is fully resolved. Each round $t$ processes an *unsolved node* $n_{target}$ through three phases:

1. **Workflow Planning (Lines 7-9):** For every specific target query $q_{target}$ (whether the original question or a derived sub-query), the Planner is invoked to generate a dedicated workflow $\mathcal{W}_t$. As illustrated in Figure 2, this step is adaptive: depending on the query's complexity, the Planner may output a Decomposition Workflow (e.g., *QDS* as shown in the top branch) to break down the problem, or construct a Solving Workflow (e.g., the chain of $RA \rightarrow DS \rightarrow AG$ as shown in the middle branch) for direct resolution.

2. **Workflow Execution (Lines 12-23):** The system executes the modules in $\mathcal{W}_t$ sequentially. If the workflow prescribes **Decomposition** (QDS/QDP), the execution expands the trace topology by appending new unsolved sub-nodes, deferring the answer to future rounds. Conversely, if the workflow prescribes **Solving**, the selected subset of executors operate in sequence to resolve the sub-problem and populate the answer slot ($a_{target}$) of the current node.

3. **State Update (Lines 25-26):** The global state $s_t$ is updated with the modified trace (either expanded with new nodes or updated with a new answer), preparing the system for the next iteration.

This process naturally adapts to complexity: simple queries use a single "Plan-Solve" iteration, while complex ones trigger a recursive "Plan-Decompose-Solve" loop.

---

[3]The **Retrieval Agent (RA)** in Table 1 is fundamentally a retriever, not an LLM; thus, we do not optimize its parameters.

[4]See Appendix F for case studies of this inference process.

*Table 1.* Definitions of Agentic Roles in JADE. The system comprises a central Planner and seven specialized Executors, categorized by their operational impact on the execution trace.

| Agent | Function Description |
|---|---|
| **Orchestration (Planner)** | |
| Planner | Monitors the current target query $q_{target}$ and orchestrates the workflow to maximize utility. |
| **Execution: Decomposition (Expands Trace with New Nodes)** | |
| *Agents in this category generate new pending sub-queries (as defined in Eq. 2) to be solved in future rounds.* | |
| QDS | **Query Decomposition (Serial).** Decomposes the question into a sequence of logically dependent sub-questions that must be resolved strictly in order. |
| QDP | **Query Decomposition (Parallel).** Decomposes the question into independent sub-questions suitable for parallel processing. |
| **Execution: Solving (Resolves Target Node)** | |
| *Agents in this category execute specific operations to resolve the target query $q_{target}$.* | |
| QR | **Query Rewriter.** Reformulates a raw question into a representation optimized for search engines. |
| RA† | **Retrieval Agent.** Fetches candidate documents relevant to the current query from external engines. |
| DS | **Document Selector.** Filters candidate documents to retain only those conducive to answering the query. |
| AG | **Answer Generator.** Generates a specific answer based on the provided evidence context. |
| AS | **Answer Summarizer.** Synthesizes the final answer to $Q_{origin}$ based on the execution trace. |

† *The Retrieval Agent functions as an interface to a frozen external retriever and is not updated; all other roles are trainable personas initialized from the LLM backbone.*

## 3.3. Reward Function

We design a hybrid reward structure that combines a *global shared reward* to foster cooperation and a *local individual penalty* to enforce structural constraints. Let a generated trajectory $\tau$ consist of a sequence of agent steps indexed by rounds $t = 1 \ldots T$ and inner steps $k = 0 \ldots K_t$.

**Global Shared Reward ($R_{global}$).** Since the Planner and Executors function as a collaborative team, the ultimate success of the task depends on their joint efforts. We define a global reward that serves as a shared feedback signal for the entire team, computed at the end of the trajectory. This signal is composed of the performance outcome and the global execution cost:

$$R_{global} = R_{perf} - \underbrace{(\alpha \cdot \mathcal{N}_{rnd}(T) + \beta \cdot \mathcal{N}_{ret}(N_{ret}))}_{R_{cost}} \quad (6)$$

where $R_{perf} = F_1(\hat{y}, y)$ measures the final answer quality. The penalty term $R_{cost}$ is explicitly defined as the weighted sum of normalized computational overheads: $T$ denotes the total number of reasoning rounds (penalizing long chains), and $N_{ret}$ denotes the total number of retrieval actions across all rounds (penalizing resource consumption). We employ linear normalization to scale these costs to $[0, 1]$ by dividing by the pre-defined maximum limit (set to 3). Controlled by coefficients $\alpha$ and $\beta$, this shared reward signal addresses the *temporal credit assignment problem*, aligning all agents in the sequence toward high-quality and efficient reasoning.

**Local Format Penalty ($r_{format}^{(t,k)}$).** While the task outcome is a collective responsibility, adherence to the output schema is an individual responsibility. If the agent active at round $t$, step $k$ generates an output that violates the required format (e.g., a Planner failing to output a valid graph topology), it receives an immediate local penalty:

$$r_{format}^{(t,k)} = \begin{cases} -1 & \text{if agent at } (t, k) \text{ violates constraints} \\ 0 & \text{otherwise} \end{cases} \quad (7)$$

**Total Reward Signal ($r_{t,k}$).** The actual reward signal $r_{t,k}$ assigned to step $(t, k)$ for optimization combines immediate behavior compliance with the team's long-term success:

$$r_{t,k} = r_{format}^{(t,k)} + R_{global}. \quad (8)$$

Here, $R_{global}$ is assigned only at the terminal step of the workflow and is zero otherwise. During the PPO update via Generalized Advantage Estimation, this terminal global reward is propagated backward to all agents in the workflow. This enables earlier agents, such as a Planner at $t = 1$, to receive credit for facilitating a successful final answer, while the local penalty $r_{format}^{(t,k)}$ provides immediate, step-specific feedback for correcting syntactic errors.

## 3.4. Joint Optimization via PPO

To bridge the strategic-operational gap, JADE optimizes the shared parameters $\theta$ using Proximal Policy Optimization (PPO) (Schulman et al., 2017). As illustrated in Figure 2 (Right), our training paradigm is designed to handle the structural complexity of dynamic agentic workflows. Unlike standard RL where an agent interacts with a uniform environment, JADE involves multiple specialized roles (Planner and Executors) generating heterogeneous data streams within a single reasoning trajectory.

To address this, we introduce a **Unified Experience Replay** mechanism. As detailed in Algorithm 2 (Appendix B), during the inference of a query batch, every agent's interaction—whether it is a Planner determining the workflow topology or a Document Selector filtering documents—is treated as a standard decision step. These heterogeneous transitions are captured, flattened, and aggregated into a shared Experience Buffer. This allows the optimization step to strictly follow the standard PPO protocol, updating the shared backbone to simultaneously improve strategic planning and operational execution.

**Heterogeneous Transition Aggregation.** As depicted in the "Buffer" component of Figure 2, the core of our optimization is the aggregation of diverse experiences. For a given batch of queries, the system performs inference in parallel. Since the workflow is dynamic, Query A might trigger a short "Planner → Retrieval" chain, while Query B

*Table 2.* Main performance comparison (F1 Score). All methods utilize Qwen2.5-7B-Instruct as the backbone for fair comparison. Best results are **bolded**, second best underlined. "Impv. vs Best" shows gain (blue) or drop (gray) against the best baseline.

| Method | Single-hop QA | | | | Multi-hop QA | | | | | Avg All |
|---|---|---|---|---|---|---|---|---|---|---|
| | NQ | PopQA | AmbigQA | Avg | HotpotQA | 2Wiki | Musique | Bam. | Avg | |
| *Standard Baselines* | | | | | | | | | | |
| LLM w/o RAG | 17.53 | 14.93 | 23.53 | 18.66 | 17.76 | 22.58 | 8.58 | 17.14 | 16.52 | 17.44 |
| Vanilla RAG | 40.60 | 42.74 | 56.20 | 46.51 | 28.33 | 25.91 | 25.20 | 25.28 | 26.18 | 34.89 |
| *RL-Based (Static Modular Workflow)* | | | | | | | | | | |
| RRR (Ma et al., 2023) | 54.60 | **50.46** | 65.41 | 56.82 | 46.21 | 41.52 | 18.27 | 36.59 | 35.65 | 44.72 |
| BGM (Ke et al., 2024) | 54.21 | 49.51 | 65.97 | 56.56 | 46.85 | 37.79 | 17.55 | 37.38 | 34.89 | 44.18 |
| MMOA-RAG (Chen et al., 2025a) | 55.44 | 50.21 | 68.02 | 57.89 | 49.21 | 41.66 | 17.26 | 37.20 | 36.33 | 45.57 |
| *Agentic Search (Adaptive Workflow)* | | | | | | | | | | |
| Adaptive RAG (Jeong et al., 2024) | 36.52 | 35.59 | 45.32 | 39.14 | 42.38 | 39.62 | 25.48 | 34.85 | 35.58 | 37.11 |
| Search-R1 (Jin et al., 2025) | 52.57 | 46.98 | 65.25 | 54.93 | 46.87 | 39.03 | 17.97 | 38.69 | 35.64 | 43.91 |
| MAO-ARAG (Chen et al., 2025b) | 36.82 | 41.85 | 47.03 | 41.90 | 46.65 | 43.96 | 22.38 | 49.84 | 40.71 | 41.22 |
| **JADE (Ours)** | **59.45** | 50.20 | **68.94** | **59.53** | **57.02** | **53.87** | **29.26** | **58.26** | **49.60** | **53.86** |
| *Impv. vs Best* | +4.01 | -0.26 | +0.92 | +1.64 | +7.81 | +9.91 | +3.78 | +8.42 | +8.89 | +8.29 |

triggers a long "Planner → Decompose → Solve" chain. Despite this structural variance, we decompose every operation into a standardized atomic transition tuple $\langle o_{t,k}, a_{t,k}, r_{t,k} \rangle$. These transitions are collected into the unified buffer $\mathcal{M}$. Crucially, $\mathcal{M}$ contains a *mixture* of role data: a sample batch sampled from $\mathcal{M}$ may simultaneously contain a high-level orchestration action from a Planner and a low-level filtering action from a Document Selector. Optimizing the shared parameters $\theta$ across this diverse mixture fosters a unified representation that effectively bridges the gap between high-level strategic planning and low-level operational execution.

**Temporal Credit Assignment via Cooperative Game.** Our optimization formulation treats the multi-turn search process as a **fully cooperative multi-agent game**. Although the Planner and Executors have distinct roles, they are bound by a *shared objective*: the global reward $R_{\text{global}}$. By propagating this collective payoff backward through the decision chain, we enforce **mutual alignment**: the Planner is incentivized to generate workflows not just for syntactic correctness, but for their executability by downstream agents, while Executors are motivated to maximize the team's final success based on the Planner's context. This mechanism solves the credit assignment problem, transforming individual greedy actions into cooperative team behaviors.

## 4. Experiments

To comprehensively evaluate the effectiveness of our proposed framework, we design our experiments to answer the following research questions:

- **RQ1: Overall Performance.** Can JADE, by unifying dynamic workflow planning with cooperative multi-agent execution, outperform state-of-the-art baselines?
- **RQ2: Efficiency-Performance Trade-off.** Can we flexibly balance effectiveness and efficiency by adjusting the

resource penalty coefficients?
- **RQ3: Ablation on Joint Optimization.** What are the specific benefits of the joint optimization strategy?

Due to space constraints, we provide the analysis for **RQ4: Multi-Agent Training Dynamics** (investigating how agents evolve to collaborate) in **Appendix D**.

**Datasets.** We assess the versatility and robustness of our framework across a comprehensive suite of open-domain QA benchmarks, stratified by reasoning complexity. For **Single-hop QA**, which primarily tests precise factual retrieval, we utilize Natural Questions (NQ) (Kwiatkowski et al., 2019), PopQA (Mallen et al., 2022), and AmbigQA (Min et al., 2020). For **Multi-hop QA**, to rigorously evaluate the capability for trajectory planning and complex reasoning, we employ HotpotQA (Yang et al., 2018), 2Wiki-MultiHopQA (Ho et al., 2020), Musique (Trivedi et al., 2022), and Bamboogle (Press et al., 2022).

**Baselines.** We benchmark JADE against three distinct categories of paradigms: **Standard Baselines**, **RL-Based Static Workflows**, and **Agentic Search Baselines**. A detailed breakdown of the implementation configurations for these methods is provided in Appendix C.

**Implementation Details.** Our training framework is built upon the official `verl` library[5], optimized for efficient RLHF. Unless otherwise stated, we employ `Qwen2.5-7B-Instruct` (Team, 2024) as the backbone LLM for all components. For retrieval, we utilize the English Wikipedia corpus, indexed via **E5** (Wang et al., 2022) to ensure high-quality dense retrieval. Performance is evaluated using the standard **F1 Score** metric.

---

[5] https://github.com/volcengine/verl

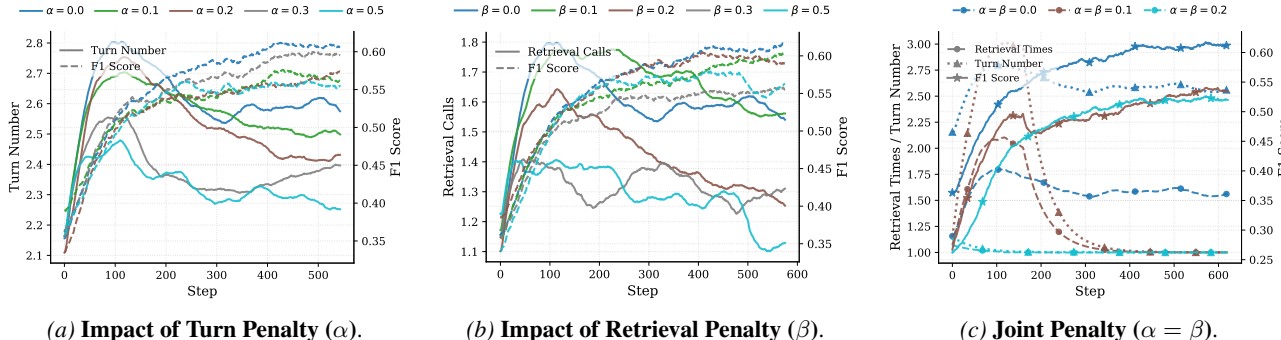

*(a)* **Impact of Turn Penalty ($\alpha$).**  *(b)* **Impact of Retrieval Penalty ($\beta$).**  *(c)* **Joint Penalty ($\alpha = \beta$).**

*Figure 3.* **Hyperparameter Sensitivity Analysis of the Cost Reward $R_{cost}$.** We analyze the training dynamics under varying penalty coefficients. **(a)** Varying the turn penalty $\alpha$ (with $\beta = 0$) effectively constrains the Planner to reduce reasoning steps (Solid Lines) while maintaining F1 Score. **(b)** Varying the retrieval penalty $\beta$ (with $\alpha = 0$) encourages Executors to reduce retrieval calls. **(c)** Jointly increasing both parameters ($\alpha = \beta$) leads to over-penalization, causing the system to rapidly degenerate into a static single-turn RAG workflow to minimize costs, resulting in pronounced performance degradation.

## 4.1. RQ1: Overall Performance Analysis

The main performance results on seven open-domain QA benchmarks are presented in Table 2. Overall, JADE achieves a new state-of-the-art performance, recording an average F1 score of **53.86** across all datasets and outperforming the previous best baseline by a remarkable margin of **+8.29**. To answer **RQ1**, we analyze the performance gains from three distinct perspectives:

**Superiority over Static Modular Workflows.** JADE demonstrates a substantial advantage over RL-based static approaches. While methods like MMOA-RAG (Chen et al., 2025a) attempt to optimize fixed pipelines (Rewriter $\rightarrow$ Selector $\rightarrow$ Generator) via multi-agent RL, they are fundamentally limited by their rigid topology. As shown in Table 2, JADE surpasses the best static baseline (MMOA-RAG) by **8.29** points on average. The gap is particularly pronounced in Multi-hop QA tasks (e.g., **+9.91** on 2Wiki, **+7.81** on Hot-potQA), confirming that the dynamic topology capability of JADE is essential for solving complex reasoning problems that static graphs cannot adequately model.

**Benefit of Joint Optimization (vs. MAO-ARAG).** A critical comparison lies between JADE and MAO-ARAG (Chen et al., 2025b). Both frameworks share a similar hierarchical architecture, employing a Planner to organize Executors for dynamic workflows. However, MAO-ARAG adopts a decoupled training strategy where only the Planner is optimized while Executors remain frozen. Our results show that JADE outperforms MAO-ARAG by an impressive **12.64** points on average (**53.86** vs. **41.22**). This substantial gap validates our hypothesis regarding the "strategic-operational mismatch." In MAO-ARAG, even if the Planner devises an optimal workflow, the frozen Executors often fail to execute the specific sub-tasks accurately, leading to system-wide failure. By contrast, JADE employs *Joint Agentic Dynamic Optimization*, enabling the Executors to co-adapt with the

Planner. This ensures that the Executors evolve to meet the specific requirements of the planned workflow, substantially mitigating execution failures.

**Advantage of Functional Specialization (vs. Search-R1).** JADE outperforms the monolithic Search-R1 (Jin et al., 2025) (**53.86** vs. **43.91**). While Search-R1's "jack-of-all-trades" design imposes excessive cognitive burden that often leads to hallucinations or reasoning drift , JADE decomposes the complex search process into specialized, atomic rolesd. This modular architecture acts as a structural scaffold to reduce the exploration space by allowing each agent to focus on simplified sub-tasks. Such *functional specialization* provides the necessary grounding to maintain precision and stability in complex scenarios where unconstrained monolithic models falter.

## 4.2. RQ2: Efficiency-Performance Trade-off

To answer **RQ2**, we investigate the system's ability to navigate the trade-off between task performance (F1) and computational cost (reasoning turns $N_{turn}$ and retrieval calls $N_{ret}$). This balance is explicitly controlled by the penalty coefficients $\alpha$ and $\beta$ in our global reward formulation (Eq. 6). Figure 3 illustrates the training dynamics under varying penalty configurations.

**Controllable Balance via Individual Penalties.** Figures 3(a) and 3(b) demonstrate the independent effects of the turn penalty ($\alpha$) and retrieval penalty ($\beta$), respectively. We observe a clear regularization pattern: as the penalty weight increases, the Planner learns to prune the workflow, resulting in a consistent reduction in the corresponding cost metric (Turn Number or Retrieval Calls). Crucially, while aggressive penalization (e.g., $\alpha = 0.5$) leads to a moderate decline in F1 score, the system retains the ability to solve a considerable portion of queries. This confirms that by adjusting $\alpha$ or $\beta$, JADE allows users to flexibly calibrate the model's behav-

ior, trading off marginal performance gains for considerable improvements in inference efficiency.

**Risk of Degeneration under Joint Penalties.** Figure 3(c) explores the impact of increasing both penalties ($\alpha = \beta$). The results reveal a "tipping point" leading to over-penalization. Specifically, with $\alpha = \beta = 0.1$ (brown curve), we observe a sharp collapse in trajectory length around step 100, where the policy rapidly converges to the same minimal-action mode observed in the stricter $\alpha = \beta = 0.2$ setting. This indicates that when the compounded cost becomes too high, the multi-agent system abandons complex reasoning strategies and degenerates into a static, single-turn "Retrieve-Generate" loop (effectively reverting to Vanilla RAG) to minimize negative rewards. This finding highlights the necessity of fine-grained hyperparameter tuning to prevent the system from collapsing into trivial solutions.

### 4.3. RQ3: Ablation on Joint Optimization

To answer **RQ3**, we conduct a comprehensive ablation study to isolate the benefits of our Joint Agentic Dynamic Optimization strategy. We analyze the impact from two perspectives: internal module co-adaptation (Table 3) and comparison against strong proprietary models (Table 4).

**Turning "Side Effects" into Gains via Co-adaptation.** Modern Agentic Search systems typically incorporate specialized modules (e.g., Document Selector) validated by prior literature. However, simply assembling these modules does not guarantee performance. As shown in Table 3, in the *Frozen Backbone* setting, explicitly adding the Document Selector (DS) module actually degrades performance compared to the base workflow (Avg: 41.74 → 41.13). This implies that without joint training, the introduction of the DS module merely increases system complexity and noise, acting as a "side effect" rather than an enhancement. In stark contrast, after applying JADE's MARL training, the inclusion of the DS module yields a noticeable performance boost (Avg: 57.10 → **58.24**). This reversal demonstrates the core value of joint optimization: it successfully aligns the Planner's orchestration with the Executors' capabilities, transforming a module that was initially a liability into a critical asset for the system.

**Co-adaptation Trumps Generic Intelligence (vs. GPT-4o).** To verify that the performance gains stem from multi-agent collaboration rather than just the generic abilities of backbone models, we fix the MARL-trained Planner and vary the Executor backbones (Table 4). In the *Frozen Executors* setting, using GPT-4o as the executor achieves the highest baseline performance (56.82), vastly outperforming the frozen Qwen-2.5-7B (41.13). However, JADE—which utilizes the much smaller Qwen-2.5-7B but optimizes it jointly with the Planner—achieves an average F1 score of **58.24**, surpassing even the GPT-4o-based system. This re-

*Table 3.* Ablation study on the impact of the Document Selector (DS) module before and after MARL training. All variants use Qwen-2.5-7B-Instruct as the backbone. **Key Insight:** While the DS module initially hurts performance on the frozen model, MARL training successfully adapts the executors to utilize DS, achieving the best performance.

| Configuration | NQ | HotpotQA | Average |
|---|---|---|---|
| *Before Training (Frozen Backbone)* | | | |
| Base Workflow (w/o DS) | 36.82 | 46.65 | 41.74 |
| + Document Selector (w/ DS) | 36.06 | 46.20 | 41.13 |
| *After MARL Training* | | | |
| JADE (w/o DS) | 58.00 | 56.20 | 57.10 |
| **JADE (w/ DS)** | **59.45** | **57.02** | **58.24** |

*Table 4.* Comparison with different backbones for the Executor. All methods utilize the same MARL-trained Planner (Qwen-2.5-7B). We compare trained JADE executors against strong models (GPT-4o series) used as frozen executors equipped.

| Executor Backbone | NQ | HotpotQA | Average |
|---|---|---|---|
| *Frozen Executors* | | | |
| Qwen-2.5-7B-Instruct | 36.06 | 46.20 | 41.13 |
| GPT-4o-mini | 54.50 | 53.90 | 54.20 |
| GPT-4o | 55.50 | **58.14** | 56.82 |
| *MARL-Tuned Executors* | | | |
| **JADE (Ours)** | **59.45** | 57.02 | **58.24** |

sult is pivotal. It confirms that the "Strategic-Operational Mismatch" cannot be solved merely by scaling up the intelligence of frozen executors. Instead, JADE demonstrates that a cohesive, co-adapted team of small models (7B) can outperform disjointed systems relying on giant proprietary models, offering superior systemic utility with a significantly better cost-performance ratio.

## 5. Conclusion

In this paper, we propose JADE (Joint Agentic Dynamic Execution), a unified framework formulating dynamic Agentic RAG as a cooperative multi-agent game. JADE effectively bridges the strategic-operational gap by surpassing existing paradigms: (1) unlike static modular workflows, it enables dynamic, multi-turn topology orchestration for complex reasoning; (2) unlike decoupled agentic systems, it optimizes executors and planners jointly to resolve strategic-operational mismatches; and (3) unlike monolithic models, its functional specialization simplifies the task for the LLM at each step, stabilizing training and enhancing performance. By utilizing shared-parameter optimization, JADE facilitates deep co-adaptation where the planner respects execution boundaries and executors evolve to fulfill strategic intent. Empirical results across seven benchmarks validate the efficacy of this cooperative paradigm, demonstrating that JADE achieves a superior balance of dynamic flexibility and

task effectiveness. Notably, this confirms that a synergistic team of smaller models can outperform disjoint systems relying on giant proprietary models.

## Acknowledgments

This research was sponsored by the Beijing Nova Program, the Natural Science Foundation of China (62572475, 61902209, 62377044), the Beijing Outstanding Young Scientist Program (NO. BJWZYJH012019100020098), the Intelligent Social Governance Platform, and the Major Innovation & Planning Interdisciplinary Platform for the "Double-First Class" Initiative, Renmin University of China.

## Impact Statement

This work studies joint optimization for dynamic agentic retrieval-augmented generation systems. Its potential positive impact lies in improving the effectiveness and efficiency of knowledge-intensive question answering, which may benefit applications such as education, research assistance, and enterprise knowledge management.

Potential risks include inheriting biases or errors from retrieved documents, amplifying outdated or incorrect information, and enabling more automated information-seeking behaviors. Therefore, such systems should be deployed with source verification, human oversight in high-stakes scenarios, and appropriate controls on external tool access and computational cost.

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

## A. Rationale for Parameter Sharing Strategy

In the JADE framework, we employ a homogeneous parameter-sharing strategy where a single Large Language Model (LLM) backbone $\pi_\theta$ serves as the underlying policy for all agentic roles (i.e., Planner, QDS, QDP, QR, DS, AG, AS), distinguished solely by role-specific system instructions. We adopt this design based on three critical considerations aligned with our goal of resolving the strategic-operational mismatch:

**1. Facilitating Co-adaptation to Bridge the Mismatch**    The core hypothesis of JADE is that decoupled optimization leads to a *Strategic-Operational Mismatch*, where the Planner's strategies drift away from the Executors' actual capabilities. Parameter sharing serves as a structural regularization that forces **co-adaptation**. By optimizing a single set of parameters $\theta$ on the aggregated experiences of both planning and execution, the gradients generated by the Executors (e.g., failure to find documents) directly update the shared representation used by the Planner. This ensures that the Planner implicitly "learns" the capability boundaries of the Executors, while Executors evolve to better interpret the Planner's strategic intent. This deep alignment is difficult to achieve with disparate policy networks.

**2. Deployment Efficiency for Complex Teams**    JADE orchestrates a sophisticated team comprising at least eight distinct functional roles. Unlike traditional RL agents based on lightweight MLPs, our agents are initialized with LLMs containing billions of parameters (e.g., Qwen-2.5-7B). Maintaining independent policy networks for every role would result in a linear scaling of memory consumption ($\mathcal{O}(N)$), making the system computationally prohibitive to train and impossible to deploy in real-world resource-constrained environments. Parameter sharing reduces the storage requirement to $\mathcal{O}(1)$, allowing us to deploy a versatile multi-agent system with the footprint of a single model. This efficiency enables us to allocate limited GPU memory toward larger batch sizes, which are crucial for stable PPO training.

**3. Latent Skill Synergy via Multi-Task Learning**    LLMs inherently possess strong multi-task capabilities, enabling them to switch roles based on contextual prompts without modifying internal weights. In JADE, seemingly distinct tasks share fundamental reasoning competencies. For instance, the *Document Selector (DS)* learns to discriminate relevant evidence from noise, a skill that shares latent representations with the *Answer Generator (AG)*, which must synthesize that evidence into a coherent response. Parameter sharing leverages this positive transfer: optimizing the backbone for data selection can implicitly enhance the reading comprehension capabilities required for answer generation. By conditioning the shared $\pi_\theta$ on role-specific prompts, we effectively project the model's general capabilities into specific functional subspaces, achieving role specialization without architectural redundancy.

**4. Alignment with Established MARL Paradigms**    Parameter sharing is not an ad-hoc design but a cornerstone strategy in the Multi-Agent Reinforcement Learning (MARL) community. Representative algorithms such as QMIX (Rashid et al., 2020), and MAPPO (Yu et al., 2022) extensively utilize parameter sharing to handle large state-action spaces and promote knowledge transfer between homogeneous or heterogeneous agents. By adopting this standard paradigm, JADE inherits the benefits of improved sample efficiency and training stability that have been rigorously validated in the broader MARL literature.

**5. Consistency with State-of-the-art Agentic Architectures**    Our design aligns with recent advancements in the specific domain of Agentic RAG and Reasoning. Existing work such as **MMOA-RAG** (Chen et al., 2025a) explicitly validates the effectiveness of parameter sharing for optimizing static modular workflows. Furthermore, even monolithic models like **Search-R1** (Jin et al., 2025) provide empirical support for this philosophy. Although Search-R1 is not explicitly modular, it tasks a single model with executing a complex sequence of cognitive operations—reasoning, search query generation, information synthesis, and answer generation—within a single context stream. This confirms that a single LLM backbone has the sufficient capacity to house the full spectrum of diverse functional capabilities required by JADE's multi-agent team.

## B. Detailed Implementation Algorithms

In this section, we provide the detailed pseudocode for the JADE framework to facilitate reproducibility. **Algorithm 1** formalizes the *Inference Workflow*, illustrating the iterative interaction between the Planner and Executors during multi-turn reasoning. **Algorithm 2** outlines the *Joint Optimization Procedure*, detailing how heterogeneous transitions from different agentic roles are aggregated into a Unified Experience Buffer for end-to-end PPO training.

**Algorithm 1** JADE Inference Workflow

---

1: **Input:** User Query $Q_{origin}$, Policy $\pi_\theta$
2: **Initialize:** Trace $\mathcal{T}_0 \leftarrow \{n_{root}\}$ where $n_{root} = \langle Q_{origin}, \emptyset \rangle$
3: **Initialize:** State $s_0 \leftarrow \{Q_{origin}, \mathcal{T}_0\}$, Round $t \leftarrow 0$
4: **while** $\exists n_m \in \mathcal{T}_t$ such that $a_m = \emptyset$ **do**
5:     $t \leftarrow t + 1$
6:     Select the first unsolved node $n_{target} = \langle q_{target}, \emptyset \rangle$ from $\mathcal{T}_{t-1}$
7:     // Phase 1: Planner orchestrates workflow (Step $k = 0$)
8:     Observe context $o_{t,0} = \mathcal{O}(q_{target}, \emptyset, \mathcal{A}_{plan})$
9:     $\mathcal{W}_t \leftarrow \text{PlanWorkflow}(\pi_\theta, o_{t,0})$ {Generate executor graph}
10:     // Phase 2: Execute the organized workflow (Steps $k = 1 \ldots K_t$)
11:     Initialize intra-round context $\text{Context}_{t,1} \leftarrow \emptyset$
12:     **for** each module $\rho_k$ in topological order of $\mathcal{W}_t$ **do**
13:         Observe $o_{t,k} = \mathcal{O}(q_{target}, \text{Context}_{t,k}, \rho_k)$
14:         Execute action $a_{t,k} \sim \pi_\theta(\cdot | o_{t,k})$
15:         **if** $\rho_k \in \{\mathcal{A}_{QDS}, \mathcal{A}_{QDP}\}$ **then**
16:             Expand $\mathcal{T}_t$ with new sub-nodes based on $a_{t,k}$
17:             **break** {Decomposition ends the round for this node}
18:         **else if** $\rho_k$ is Solving Agent (QR, RA, DS, AG) **then**
19:             Update $\text{Context}_{t,k+1} \leftarrow \text{Context}_{t,k} \cup \{a_{t,k}\}$
20:             **if** $\rho_k == \mathcal{A}_{AG}$ **then**
21:                 Update node $n_{target}$ with answer $a_{target} \leftarrow a_{t,k}$
22:             **end if**
23:         **end if**
24:     **end for**
25:     // Phase 3: Update Global State
26:     $s_t \leftarrow \{Q_{origin}, \mathcal{T}_t\}$
27:     **if** Max steps reached **or** $\forall n_m \in \mathcal{T}_t, a_m \neq \emptyset$ **then**
28:         **break**
29:     **end if**
30: **end while**
31: // Final Synthesis
32: predicted_answer $\leftarrow \mathcal{A}_{AS}(\mathcal{T}_t)$
33: **return** predicted_answer

---

## C. Implementation Details of Baselines

To validate the effectiveness of JADE, we compare it against a diverse set of state-of-the-art methods categorized as follows:

- **Standard Baselines:** *LLM w/o RAG* (Parametric knowledge only) and *Vanilla RAG* (Standard retrieval-generation).

- **RL-Based (Static Modular Workflow):** *RRR* (Ma et al., 2023) (Query rewriting optimization), *BGM* (Ke et al., 2024) (Document selection optimization), and *MMOA-RAG* (Chen et al., 2025a) (Joint optimization of fixed modules).

- **Agentic Search (Adaptive Workflow):** *Adaptive RAG* (Jeong et al., 2024) (Complexity-based routing), *Search-R1* (Jin et al., 2025) (End-to-end reasoning agent), and *MAO-ARAG* (Chen et al., 2025b) (Planner-centric optimization).

To ensure a fair and rigorous comparison, we unify the experimental setting across all baselines and JADE. We utilize `Qwen2.5-7B-Instruct` as the consistent backbone model. For baseline components that do not require training, we use the original pre-trained checkpoint. For trainable components, we initialize them with `Qwen2.5-7B-Instruct` and perform fine-tuning or RL training as specified by their respective methodologies. Specific implementation notes are detailed below:

**Standard Baselines**

---

**Algorithm 2** JADE Joint Optimization Procedure

---

1: **Input:** Training Dataset $\mathcal{D}_{train}$, LLM Policy $\pi_\theta$, Value Network $V_\phi$
2: **Hyperparameters:** Learning rate $\eta$, Batch size $B$, Clip $\epsilon$, GAE parameters
3: **for** epoch $= 1, \ldots, E$ **do**
4:     Shuffle $\mathcal{D}_{train}$ and split into batches
5:     **for** each batch $\mathcal{B} = \{Q_1, \ldots, Q_B\} \subseteq \mathcal{D}_{train}$ **do**
6:         Initialize Experience Buffer $\mathcal{M} \leftarrow \emptyset$
7:         // Phase 1: Batch Inference & Data Collection (See Fig. 2 Left)
8:         **Parallel For** each query $Q_j \in \mathcal{B}$:
9:             $\tau_j \leftarrow$ JADE_Inference$(Q_j, \pi_\theta)$ {Generates hierarchical trace}
10:           Compute terminal global reward $R_{\text{global}}$ (Eq. 6)
11:           // Flatten hierarchical rounds ($t$) and steps ($k$) into linear sequence
12:           Flatten $\tau_j \rightarrow \langle(o_{0,0}, a_{0,0}), \ldots, (o_{T,K}, a_{T,K})\rangle$
13:           Compute advantages $\hat{A}_{t,k}$ using GAE over the flattened sequence
14:           **For** each transition in flattened $\tau_j$:
15:              $\mathcal{M}$.push($\langle o_{t,k}, a_{t,k}, r_{t,k}, \hat{A}_{t,k}, \log \pi_{old}\rangle$)
16:         **End Parallel For**
17:         // Phase 2: Joint Update from Unified Buffer (See Fig. 2 Right)
18:         **for** $step = 1, \ldots, N_{opt}$ **do**
19:           Sample mixed mini-batches $b \sim \mathcal{M}$
20:           Update $\theta$ via $\nabla_\theta \mathcal{L}^{PPO}(\theta)$
21:           Update $\phi$ via $\nabla_\phi \mathcal{L}^{Val}(\phi)$
22:         **end for**
23:     **end for**
24: **end for**

---

- **LLM w/o RAG:** A closed-book baseline where the model generates answers relying solely on its pre-trained parametric memory, without any external retrieval.

- **Vanilla RAG:** The canonical retrieve-then-generate pipeline. It retrieves the top-5 documents using the original query and feeds them into the pre-trained LLM for direct answer generation.

**Static Modular Workflow (RL-Based)**

- **RRR** (Ma et al., 2023): A framework that trains a Query Rewriter using PPO. Following the robust reproduction protocols, we strictly align the reward signals and fine-tune the subsequent generator to prevent capabilities mismatch.

- **BGM** (Ke et al., 2024): This method optimizes a Bridge Module (document selector) via reinforcement learning to bridge the gap between retrieval and generation. We reproduce this by training the selector to maximize the generator's likelihood of the ground truth.

- **MMOA-RAG** (Chen et al., 2025a): Represents the state-of-the-art in static joint optimization. It employs Multi-Agent PPO to simultaneously train the Query Rewriter, Document Selector, and Generator. We strictly follow its fixed-graph topology (Rewriter $\rightarrow$ Selector $\rightarrow$ Generator) using the shared backbone.

## D. Analysis of Multi-Agent Training Dynamics (RQ4)

In this section, we address **RQ4** by visualizing the emergent behavioral patterns of the agents during the training process. Figure 4 tracks the evolution of workflow strategies and module utilization across Single-Hop (NQ) and Multi-Hop (HotpotQA) tasks.

**Evolution of Adaptive Workflows.** Figures 4(a) and 4(b) illustrate the distribution of workflow types chosen by the Planner. On the Single-Hop task (Figure 4(a)), we observe that while the Planner initially explores decomposition strategies (QDS/QDP), it rapidly converges to a dominance of *Single-Round* workflows. This indicates that the Planner correctly

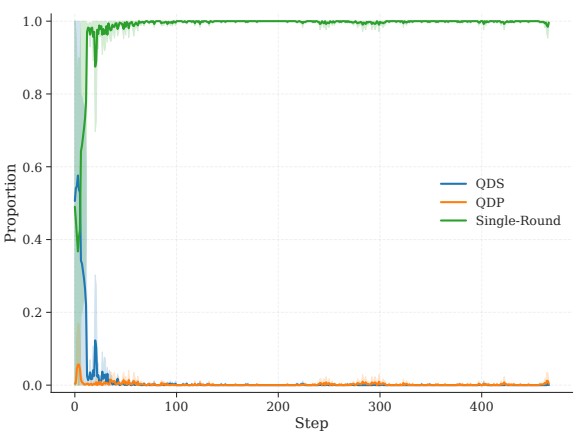

*(a)* **Evolution of Workflow Strategies on Single-Hop QA (NQ).** The Planner rapidly converges to Single-Round workflows, optimizing for efficiency.

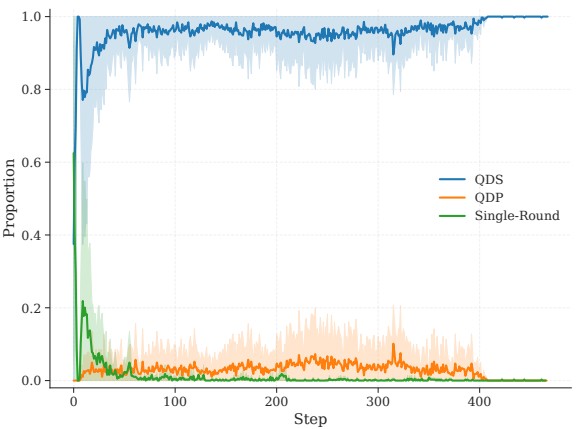

*(b)* **Evolution of Workflow Strategies on Multi-Hop QA (HotpotQA).** The Planner learns to favor Serial Decomposition (QDS) over Parallel (QDP) to handle context dependencies.

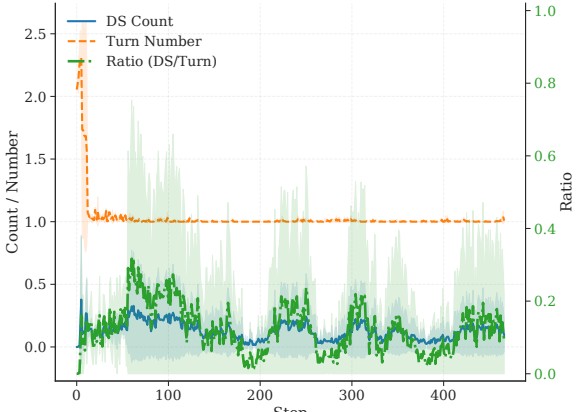

*(c)* **Document Selector (DS) Usage on Single-Hop QA.** The low utilization ratio ($< 0.2$) indicates that filtering is largely unnecessary for simple queries.

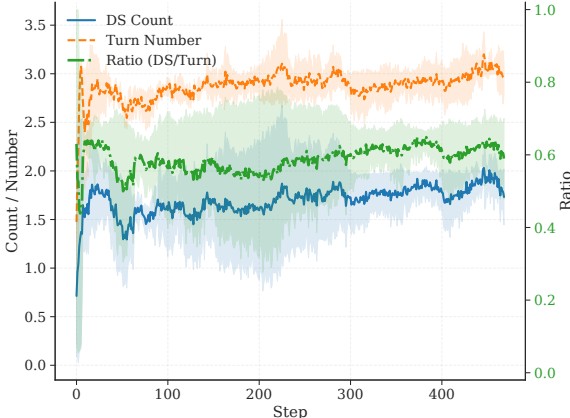

*(d)* **Document Selector (DS) Usage on Multi-Hop QA.** The high utilization ratio ($\approx 0.6$) confirms that active noise filtering is critical for complex reasoning.

*Figure 4.* **Detailed Analysis of Workflow Patterns and Document Selector (DS) Dynamics. (a)** and **(b)** illustrate how the Planner's strategy evolves over training steps across different task complexities. **(c)** and **(d)** depict the utilization intensity of the Document Selector, calculated as the ratio of DS calls to total reasoning turns.

identifies that simple factual queries do not require complex decomposition, thereby optimizing for efficiency. Conversely, on the Multi-Hop task (Figure 4(b)), the system trends towards *Query Decomposition (Serial)* (QDS). Interestingly, while *Query Decomposition (Parallel)* (QDP) appears in the early-to-mid training stages, QDS eventually becomes the dominant strategy. This convergence suggests that for complex reasoning chains, the sequential dependency between sub-queries is a hard constraint that conceptually overrides the efficiency of parallelism. Although QDS incurs higher latency due to serial execution, the Planner learns that accessing prior answers is prerequisite for formulating valid subsequent queries. Consequently, the system autonomously navigates the trade-off, prioritizing the robustness of context-aware reasoning over the computational parallelism of QDP.

**Strategic Utilization of Document Selector.** Figures 4(c) and 4(d) monitor the utilization intensity of the Document Selector (DS) relative to the total reasoning turns. For Single-Hop QA (Figure 4(c)), the trajectory length converges to 1, with a low DS/Turn ratio ($< 0.2$). This implies that for simple queries, the system learns that filtering is often unnecessary overhead, preferring to ingest retrieved documents directly. In contrast, for Multi-Hop QA (Figure 4(d)), the system stabilizes at approximately 3 turns, with a high DS utilization ratio ($\approx 0.6$). This indicates that in more than half of the reasoning steps,

the agents actively choose to filter noise. These distinct behaviors are not hard-coded but are learned emergent properties of the collaborative Planner-Executor optimization, where the team autonomously adjusts its "cognitive" effort based on task complexity.

## E. Agent Prompts

### 1. Planning Agent

You are a helpful assistant specialized in planning workflows. Your task is to plan a Workflow for the given question using the available tools/agents.

Available Tools/Agents:

- Query Rewriter (QR): Input: question → Output: rewritten question that is more concise, clearer, and accurate.

- Query Decomposition Serial (QDS): Input: question → Output: dependent sub-questions where later ones depend on earlier ones.

- Query Decomposition Parallel (QDP): Input: question → Output: several sub-questions can be searched independently.

- Retrieval (R): Input: question → Output: relevant candidate documents.

- Document Selector (DS): Input: question + candidate documents → Output: subset of documents helpful for answering.

- Answer Generator (AG): Input: question [+ optional documents] → Output: final answer.

Rules for tool selection:

1. When the question needs to be broken down into sub-questions:
   - If sub-questions have dependencies and must be answered in sequence, use QDS or QDP ONLY.

2. When the question can be answered directly without decomposition:
   - Build workflow from QR, R, DS, AG.
   - If DS is in workflow, R must appear before DS.
   - The last module must always be AG.

3. IMPORTANT:
   - If you choose QDS or QDP, DO NOT add any other tools/agents.
   - The workflow must ONLY contain QDS or QDP in those cases.

Question: {query}

Now, generate the appropriate Workflow based on the rules.

Output strictly inside `<workflow>...</workflow>` tags.

### 2. Query Rewrite Agent

You are a professional assistant skilled at rewriting slightly redundant or overly wordy factual questions into a single, concise, and searchable query.

Task Requirements:

- Keep all essential names, dates, and terms.

- Do not add explanations or unrelated details.

- Make the query short and clear.

Original Question: {query}

Now, rewrite the original question. Output strictly inside `<query>...</query>` tags.

---

### 3. Query Decomposition Agent (Parallel)

You are a professional assistant skilled at decomposing complex multi-entity or multi-location questions into multiple independent sub-questions.

Task Requirements:

- Each sub-question must be specific, logically complete, and searchable independently.

- Avoid duplication and overlap.

- Do not generate more than 4 sub-questions.

Original Question: {query}

Now, break down the question into independent sub-questions. Output each sub-question on a new line inside numbered tags (`<q1>...</q1>`, `<q2>...</q2>`, etc.).

---

### 4. Query Decomposition Agent (Serial)

You are a professional assistant skilled at decomposing complex questions into a minimal sequence of logically dependent sub-questions.

Task Requirements:

- Each sub-question must be self-contained and specific.

- Ensure a logical chain where later questions depend on earlier ones.

- Keep the number of sub-questions minimal (max 4).

- Avoid redundancy.

Original Question: {query}

Now, decompose the question into a logically ordered sequence. Output each sub-question on a new line inside numbered tags (`<q1>...</q1>`, `<q2>...</q2>`, etc.).

---

### 5. Document Selection Agent

You are a helpful, respectful, and honest assistant. Your task is to identify which candidate Documents are helpful in answering the Question.

Question: {query}

{doc_content}

Now, select the helpful documents. Output their IDs (0, 1, ..., {max_id}) as comma-separated values strictly inside `<id>...</id>` tags.

---

## 6. Answer Generation Agent

You are a helpful, respectful, and honest assistant. Your task is to provide a brief and accurate answer to the Question based on the provided Documents.

Task Requirements:

- Answer strictly based on the documents.

- If the answer is not in the documents, say "I don't know".

- Do not fabricate information.

Question: {query}

{doc_content}

Now, generate the brief and accurate answer. Output strictly inside `<answer>...</answer>` tags.

---

## 7. Answer Summarization Agent

You are a helpful, respectful, and honest assistant. Your task is to predict the final answer to the Original Question based on the answers to its decomposed sub-questions.

Task Requirements:

- Synthesize the information from the sub-questions and observations.

- Always provide the answer you think is most correct.

- Do not answer "I don't know" unless absolutely necessary.

Original Question: {query}

{observation}

Now, answer the Original Question based on the observations. Output strictly inside `<answer>...</answer>` tags.

---

# F. Qualitative Case Studies

In this section, we provide step-by-step case studies to qualitatively demonstrate how JADE's multi-agent team collaborates to solve queries of varying complexity.

**Case 1: Complex Multi-Hop Reasoning.** This case illustrates a multi-hop query characterized by a latent dependency, where the subject (the actor) is not explicitly named. JADE's Planning Agent correctly identifies the need for sequential reasoning and activates the **Query Decomposition (Serial)** strategy. The system successfully bridges the knowledge gap by utilizing the intermediate answer ("Fred Astaire") to formulate the subsequent heritage-related query, demonstrating the effective chaining of sub-tasks.

| Agent / Phase | Action & Output |
|---|---|
| User Input | **Query:** "Something's Gotta Give was first performed by an actor of what heritage?" |
| Planning Agent | **Output:** `<workflow>QDS</workflow>` |
| Query Decomposition Agent Serial | **Sub-queries generated:**
1. "Who is the actor that first performed Something's Gotta Give?"
2. "What is the heritage of this actor?" |
| | *Sub-task 1: Identify the Actor* |
| Query Rewrite Agent | **Rewritten Query:** "Actor who first performed Something's Gotta Give" |
| Sub Planning Agent | **Plan:** `<workflow>R,AG</workflow>` |
| Retrieval Context | *Retrieved documents regarding "Something's Gotta Give" (Song and Film).*
**Key Excerpt:** "...written for and first performed by **Fred Astaire** in the 1955 musical film 'Daddy Long Legs'..." |
| Answer Generation Agent | **Intermediate Answer:** "Fred Astaire" |
| | *Sub-task 2: Identify Heritage* |
| Query Rewrite Agent | **Rewritten Query:** "Heritage of Fred Astaire" |
| Sub Planning Agent | **Plan:** `<workflow>R,DS,AG</workflow>` |
| Retrieval Context | *Retrieved documents regarding Fred Astaire.*
**Key Excerpt:** "Fred Astaire ... was an **American** dancer, singer, actor..." |
| Document Selection Agent | **Selected IDs:** `<id>Document0, Document1, Document2, Document4</id>` |
| Answer Generation Agent | **Intermediate Answer:** "American" |
| Answer Summarization Agent | **Final Output:** `<answer>American</answer>` |
| Verification | **Golden Answer:** American
**Status:** **Correct** |

**Case 2: Efficient Single-Hop Retrieval.** In contrast to the previous example, this case demonstrates JADE's efficiency in handling explicit, single-hop fact retrieval. The Planning Agent recognizes that the query requires direct evidence lookup rather than complex reasoning, opting for a streamlined **Retrieve-Select-Generate** workflow (R → DS → AG). This highlights the system's flexibility in avoiding unnecessary computational overhead for straightforward questions.

| Agent / Phase | Action & Output |
|---|---|
| User Input | **Query:** "when did canada become fully independent from britain?" |
| Planning Agent | **Output:** `<workflow>R,DS,AG</workflow>` |
| Retrieval Context | *Retrieved documents regarding the Canada Act 1982 and the Statute of Westminster.*
**Key Excerpt:** "...Canada severed its last legal tie with the UK and became **fully independent** in **1982** when the Constitution Act was patriated..." |
| Document Selection Agent | **Selected IDs:** `<id>Document0, Document1, Document4</id>` |
| Answer Generation Agent | **Final Output:** `<answer>1982</answer>` |

| Agent / Phase | Action & Output |
|---|---|
| Verification | **Golden Answer:** 1982 
 **Status: Correct** |

