# OpenReview forum: "JADE: Bridging the Strategic-Operational Gap in Dynamic Agentic RAG"
_ICML.cc/2026/Conference — ICML 2026 regular_

### Official Review · Reviewer_63Gf · 2026-02-28

**Soundness:** 3
**Presentation:** 3
**Significance:** 3
**Originality:** 3
**Overall Recommendation:** 5
**Confidence:** 3

**Summary:**

This paper proposes JADE, a shared-parameter multi-agent framework for dynamic Agentic RAG. The authors identify a "strategic–operational mismatch" between planners and frozen executors in prior systems, and address it via joint MARL optimization using PPO. Empirically, JADE shows consistent gains over static modular pipelines, decoupled planner-only RL, and monolithic RL baselines.

**Compliance With Llm Reviewing Policy:**

Affirmed.

**Key Questions For Authors:**

1. What dataset is used for training, and how large is it?

**Limitations:**

See "Strengths And Weaknesses".

**Strengths And Weaknesses:**

**Strengths**

1. The paper identifies a clear and practically relevant limitation in existing Agentic RAG systems, namely the "strategic–operational mismatch" between high-level planners and frozen executors. This diagnosis is well articulated and grounded in current trends in dynamic agentic workflows. The motivation is compelling, and the proposed solution directly addresses a concrete structural issue rather than making incremental improvements to isolated components.
2. The empirical evaluation is thorough and covers multiple important baselines, including static modular RL systems, decoupled planner-only optimization approaches, and monolithic RL frameworks. The ablation studies are well constructed and provide meaningful insight into the role of joint optimization and module co-adaptation.
3. Beyond reporting aggregate performance numbers, the paper includes detailed analysis of training dynamics, hyperparameter sensitivity, and efficiency–performance trade-offs. This level of analysis enhances the overall technical rigor and makes the empirical conclusions more convincing.

**Weaknesses**

1. While the application to dynamic Agentic RAG is interesting, the core technical ingredients, multi-agent RL with shared parameters and global rewards, are not fundamentally new. Similar ideas have been explored in prior work on cooperative MARL and shared-policy training, including self-play and parameter-sharing frameworks. As such, the primary contribution lies in the integration and application of these techniques to agentic workflows, rather than in introducing new RL algorithms or theoretical insights.

2. The experiments are conducted on standard RAG-style QA datasets (e.g., NQ, HotpotQA), which, while widely used, are relatively mature benchmarks. It would strengthen the paper to evaluate the proposed framework on more recent and challenging settings, such as open-ended deep research tasks. Demonstrating gains in more complex, modern agentic scenarios would better establish the broader impact and robustness of the approach.

---

> ### Author Rebuttal · Authors · 2026-03-30
>
> # Response to W1
> Thank you for this thoughtful and accurate assessment. We agree that the core RL ingredients in JADE, such as parameter sharing and global/shared rewards, are not new algorithmic contributions by themselves, and have been well studied in prior cooperative MARL work such as QMIX and MAPPO.
>
> **Our main contribution is to bring these established MARL principles into the LLM-based multi-agent setting for dynamic Agentic RAG**, where the challenges are substantially different from traditional neural MARL with MLP/RNN/GRU agents. Specifically, we show that a MARL-style formulation can effectively train a dynamic planner-executor architecture in LLM-based QA workflows, and that this joint optimization leads to strong performance across a broad set of QA benchmarks.
>
> We will revise the paper to make this contribution more precisely framed as a new application and system formulation, rather than a new RL algorithm or theoretical result.
>
> # Response to W2
> Thank you for this very insightful comment. We agree that evaluating JADE on more modern and challenging agentic settings would further strengthen the paper. In particular, open-ended deep research/search scenarios would be a valuable testbed for broader impact and robustness.
>
> At the same time, many publicly discussed deep research/search benchmarks, such as **GAIA** and **BrowseComp**, are evaluation-oriented and only provide test sets. This reflects the fact that annotating or constructing high-quality training data for such long-horizon, complex multi-hop tasks is itself extremely challenging. In practice, demonstrating strong performance on these benchmarks often requires first synthesizing training data with a similar distribution, followed by additional SFT/RL training, which is a substantial research topic on its own and beyond the scope of this paper.
>
> We fully agree with your point, and **we will highlight complex agentic data synthesis + MARL training for open-ended deep research tasks** as an important direction for future work.
>
> # Response to "What dataset is used for training, and how large is it?"
> Thank you for the question. We train JADE separately on **NQ** and **HotpotQA**. The training sets contain **79,168** queries for **NQ** and **90,447** queries for **HotpotQA**, respectively. We will clarify this in the revision.

---

> > ### Author Rebuttal · Reviewer_63Gf · 2026-04-03
> >
> > Thanks for the reply, which has adequately addressed my concerns. I will keep my positive score.

---

### Official Review · Reviewer_dBM6 · 2026-03-12

**Soundness:** 3
**Presentation:** 2
**Significance:** 3
**Originality:** 3
**Overall Recommendation:** 4
**Confidence:** 4

**Summary:**

The authors of this paper propose a shared-backbone framework, namely JADE, for jointly training multiple role-specialized agents in dynamic agentic RAG. The main idea of this paper is to model planning and execution as a cooperative multi-agent workflow, where trajectories from multiple rounds of planning and execution are collected, assigned rewards based on final answer quality, formatting constraints, retrieval cost, and reasoning depth, and then flattened into a unified PPO training pipeline. The experiments show that JADE outperforms the baselines including Search-R1 on several single-hop and multi-hop QA benchmarks.

**Compliance With Llm Reviewing Policy:**

Affirmed.

**Key Questions For Authors:**

1.Could the authors compare a strong frozen planner like GPT-4o with few-shot prompting of those cases where planner gets positive reward + trained executors against the fully trained JADE system? From implementation, it appears that the planner is mainly trained on role-specific inputs centered around the query and intermediate outputs, rather than full executor-side reasoning process. So the observed results may come from the planner learns to use task-specific patterns while the executors simply become better at solving the resulting sub-problems.

2.Can the authors isolate the contribution of planner adaptation versus executor adaptation? Like frozen planner + trained executors? I want to know whether joint synergy really beyond either component alone.

**Limitations:**

yes

**Strengths And Weaknesses:**

Strengths:
1. The experiment results are strong and consistent. The result that a 7B model outperforms GPT-4o-based systems proves collaboration matters more than model scale.
2. Low deployment cost and tunable efficiency of JADE make it practical for real-world use.
3.The paper tackles an importance problem with a coherent overall framework, which focuses on real challenges of coordinating planner with executors in multi-agent RAG systems.

Weaknesses:
1. The authors' core claim that joint optimization induces genuine planner-executor co-adaptation is not sufficiently validated by the experiments. The experiment results show that JADE performs well, but they fail to disentangle where the gains come from: planner or executor or joint synergy between the two? I suggest including role-isolation analyses such as frozen planner + trained executors.
2. All modules in JADE share the same parameters, implying that JADE can be interpreted as a single shared policy trained to perform difference roles. Some parameter-sharing ablations would be helpful.
3. It doesn't fully solve the problem that stated problem—planner and executors are still separate, while better coordinated and training could easily collapse. Performance is limited by frozen external search systems.
4. The evaluations are conducted only on academic benchmarks, including more real-world tasks would be better.
5. Given JADE's complex architecture, the paper does not compare response time or latency with other methods. It only reports accuracy scores like F1. Adding a comparison of how fast each method runs would make the evaluation more complete and practical.

---

> ### Author Rebuttal · Authors · 2026-03-30
>
> # Response to W1
> Thank you for this insightful suggestion. We agree that role-isolation analysis is important for disentangling the source of gains. The original paper already includes **Trained Planner + Frozen Executors** vs. **JADE**; following the reviewer’s suggestion, we further added **Frozen Planner + Trained Executors** with two frozen planners.
>
> | Setting | NQ | HotpotQA | Average |
> |---|---:|---:|---:|
> | Frozen Planner (**Qwen-2.5-7B-Instruct**) + JADE-trained Executors | 55.72 | 51.28 | 53.50 |
> | Frozen Planner (**GPT-4o**) + JADE-trained Executors | 55.34 | 52.60 | 53.97 |
> | JADE-trained Planner + Frozen Executors (**Qwen-2.5-7B-Instruct**) | 36.06 | 46.20 | 41.13 |
> | JADE-trained Planner + Frozen Executors (**GPT-4o-mini**) | 54.50 | 53.90 | 54.20 |
> | JADE-trained Planner + Frozen Executors (**GPT-4o**) | 55.50 | **58.14** | 56.82 |
> | **JADE (jointly trained Planner + Executors)** | **59.45** | 57.02 | **58.24** |
>
> Here, **JADE-trained executors** are jointly trained from the Qwen-2.5-7B-Instruct backbone. The results show that improving only one side helps, but is insufficient to match full JADE. **The best overall average is achieved only by jointly training both planner and executors**, which supports our claim that the gain comes from planner-executor co-adaptation under joint optimization.
>
> # Response to W2
> Thank you for this insightful comment. Without parameter sharing, there are two main alternatives: (1) assigning each agent its own full LLM, or (2) sharing the backbone but using separate LoRA adapters for different agents.
>
> The first is highly inefficient in our setting: JADE has 6 LLM-based agents, so it would require deploying 6 separate LLMs for one search task, causing substantial resource overhead and low GPU utilization.
>
> For the second, we performed an ablation with separate LoRA adapters. The results are very close to our fully shared setting:
>
> | Dataset | Full-Parameter + Shared (Ours) | Separate LoRA Adapters |
> |-|-:|-:|
> | NQ | 59.45 | 59.28 |
> | HotpotQA | 57.02 | 57.19 |
>
> This suggests that the performance difference between shared parameters and role-specific adapters is minimal in our setting, while parameter sharing is clearly simpler and more efficient. We will include this ablation and clarify this design choice in the revision.
>
> # Response to W3
> Thank you for this important comment. We would like to clarify that **our claim is not that JADE fully eliminates the structural separation between planner and executors**. Rather, JADE **substantially alleviates** the strategic-operational mismatch by jointly optimizing all agents, instead of optimizing only the planner while keeping executors fixed.
>
> In other words, the contribution of JADE is to enable planner-executor co-adaptation through joint optimization, rather than to remove their structural distinction entirely. **We will revise the wording to make this point more precise.**
>
> Regarding training stability, we additionally conducted 10 independent runs:
> | Method | @200 Steps | @500 Steps | @1000 Steps |
> |-|-:|-:|-:|
> | Search-R1 | 1/10 | 7/10| 9/10 |
> | **JADE (Ours)** | **0/10** | **0/10** | **0/10** |
>
> JADE shows 0/10 collapses at 200, 500, and 1000 steps, while Search-R1 collapses in 1/10, 7/10, and 9/10 runs, respectively. **This provides direct evidence that JADE is in fact highly stable in practice.**
>
> **We also agree that performance is still bounded by the frozen external search system, and we will explicitly acknowledge this limitation in the revision.**
>
> # Response to W4
> Thank you for this helpful suggestion. We agree that evaluating on more real-world tasks would further strengthen the paper. At the same time, we note that benchmarking on established academic datasets is still the standard practice in this literature, including for recent agentic search and agentic RAG work, as it enables controlled and reproducible comparison.
>
> We therefore chose these widely used benchmarks to ensure fairness and comparability. We agree that broader real-world evaluation is important, and we will highlight it as an important direction for future work.
>
> # Response to W5
> Thank you for this important comment. We agree that efficiency is an important practical dimension. Although the current submission mainly reports accuracy, JADE is in fact more efficient than Search-R1 in our setting. As shown below, JADE produces much shorter outputs during both training and inference, which leads to substantially higher throughput.
>
> | Metric | Search-R1 | **JADE (Ours)** |
> |-|-:|-:|
> | Avg. Output Length during Training (tokens) | 1100 ~ 1200 | **21 ~ 22** |
> | Total Output Tokens per Query (Inference) | 1100 ~ 1200 | **260 ~ 270** |
> | Throughput (Queries per Second) | 5.00 | **9.10** |
>
> These results suggest that, despite its multi-agent structure, JADE is not slower in practice; instead, its decomposed design leads to shorter generation trajectories and better serving efficiency. We will include this analysis in the revision.

---

> > ### Author Rebuttal · Reviewer_dBM6 · 2026-04-02
> >
> > Thanks for the rebuttal addressing my concerns. I will keep my original scores since they are positive.

---

### Official Review · Reviewer_7czo · 2026-03-13

**Soundness:** 3
**Presentation:** 3
**Significance:** 3
**Originality:** 2
**Overall Recommendation:** 4
**Confidence:** 4

**Summary:**

This paper proposes JADE, a framework for dynamic agentic RAG that jointly optimizes a planner and multiple executor roles using a shared LLM backbone and PPO. The key idea is to treat planning and execution as a cooperative multi-agent system, where shared parameters are intended to reduce the mismatch between high-level plans and specific executor capabilities. Empirically, the paper evaluates JADE across seven open-domain QA benchmarks and reports performance improvements over static modular baselines, planner-only agentic baselines, and a monolithic search-style baseline. The study also includes cost-sensitive reward tuning and ablation studies regarding the document selector module and the choice of executor backbones.

**Compliance With Llm Reviewing Policy:**

Affirmed.

**Final Justification:**

The authors' response has addressed my concerns. I believe the paper now meets the acceptance standard, and I have adjusted my score accordingly.

**Key Questions For Authors:**

same as weakness

**Limitations:**

yes

**Strengths And Weaknesses:**

## Strengths
* The paper identifies a concrete, impactful gap in dynamic agentic RAG: the strategic-operational mismatch between trained planners and frozen executors. It presents a clear, well-motivated research narrative that resonates with real-world system limitations.
* The work is well-presented with intuitive figures that clearly illustrate the framework taxonomy, inference loop, and shared-buffer PPO training workflow, making the core architecture easy to follow.
* The authors provide detailed experimental setups and analytical conclusions. The evaluation covers a reasonable range of open-domain QA tasks and further demonstrates the trade-off between efficiency and performance through ablation studies, resulting in a robust set of experimental findings.

## Weakness
* **Insufficient Contextualization within Multi-Agent Systems (MAS)**: While the paper introduces the JADE framework for flexible multi-agent scheduling, numerous relevant works in the MAS domain already exist (e.g., Chain-of-Agents). The manuscript does not sufficiently demonstrate JADE's superiority or distinct advantages over these existing frameworks.
* **Lack of Comparison with Recent SOTA Baselines:** The experimental comparison is primarily focused on Search-R1. However, the paper misses comparisons with more recent or concurrent "Agentic Search" models, such as Search-o1, AFM, ZeroSearch, and ReasonRAG.

---

> ### Author Rebuttal · Authors · 2026-03-30
>
> # Response to Weakness 1
> Thank you for this important comment. We agree that JADE should be better contextualized within the broader MAS literature.
>
> Compared with **general MAS frameworks such as AutoGen, MetaGPT, or CAMEL**, the key distinction of JADE is not merely the existence of multiple agents, but the optimization objective: **JADE is not only an orchestration/scheduling framework, but a joint multi-agent RL optimization framework that jointly trains all agents and explicitly addresses the strategic-operational mismatch.**
>
> Compared with the mentioned Chain-of-Agents (AFM), the difference is also substantial. AFM formulates multi-agent collaboration within a single model, by defining multiple tool/role agents in the prompt and dynamically activating them to simulate multi-agent collaboration end-to-end. In the QA setting, this still remains **closer to the ReAct + RL paradigm** represented by Search-R1, i.e., a single-model agent iteratively reasons, calls tools, and is optimized through RL over the full trajectory. By contrast, **JADE explicitly decomposes the workflow into specialized agents with their own input/output/reward, which simplifies each agent’s functional burden and enables stable joint optimization across the whole workflow.**
>
> We also additionally compared JADE against AFM-RL on the shared multi-hop QA benchmarks, using the same LLM-as-judge evaluation protocol as AFM-RL for alignment. As shown below, JADE achieves better performance on 6 datasets, which further supports the advantage of our multi-agent training paradigm.
>
> | Method | NQ | PopQA | HotpotQA | 2Wiki | Bamboogle | MuSiQue |
> |---|---:|---:|---:|---:|---:|---:|
> | AFM-RL | 43.9 | 46.5 | 43.9 | 49.2 | 49.6 | 22.3 |
> | **JADE (Ours)** | **55.2** | **47.3** | **53.7** | **50.1** | **55.4** | **26.0** |
>
> # Response to Weakness 2
>
> Thank you for this important comment. In addition to the baselines included in the main paper, we further compared JADE with several recent or concurrent agentic search / agentic RAG methods, including **Search-o1, R1-Searcher, StepSearch, ZeroSearch, ReasonRAG, and TIPS**. As shown below, JADE consistently outperforms these recent baselines on the shared benchmarks, which further strengthens the empirical support for our method. We will include this discussion in the revision.
>
> | Method | NQ | HotpotQA | 2Wiki | Bamboogle | Musique |
> |---|---:|---:|---:|---:|---:|
> | Search-o1 (EMNLP 25) | 32.4 | 44.9 | 39.7 | 37.6 | 20.1 |
> | R1-Searcher | 41.2 | 55.6 | 50.6 | 54.6 | 27.7 |
> | StepSearch (EMNLP 25) | 39.8 | 50.0 | 44.4 | 50.3 | 27.5 |
> | ZeroSearch| 47.2 | 39.6 | 41.9 | 32.5 | 20.1 |
> | ReasonRAG (NeurIPS 25) | 39.7 | 41.5 | 36.6 | 40.5 | 16.4 |
> | TIPS (ICLR 26) | 40.4 | 46.0 | 43.0 | 46.2 | 18.6 |
> | **JADE (Ours)** | **59.5** | **57.0** | **53.9** | **58.3** | **29.3** |
>
> **Overall, we hope these clarifications address the reviewer’s concerns. In summary, JADE is distinct from general MAS frameworks because its core contribution lies in joint multi-agent RL optimization, rather than orchestration alone; it is also substantively different from AFM/Chain-of-Agents in both modeling paradigm and training objective. Moreover, our additional comparisons with AFM-RL and several recent agentic search baselines further support the effectiveness and competitiveness of JADE. We will incorporate these clarifications and additional results in the revision.**

---

> > ### Author Rebuttal · Reviewer_7czo · 2026-04-06
> >
> > Thank you for the authors' response. I will adjust my score accordingly.

---

### Official Review · Reviewer_dp4Y · 2026-03-13

**Soundness:** 3
**Presentation:** 3
**Significance:** 4
**Originality:** 3
**Overall Recommendation:** 4
**Confidence:** 4

**Summary:**

This paper introduces JADE, a framework that addresses the mismatch between planning strategies and execution in dynamic agentic RAG systems. The authors argue that existing systems treat executors as fixed tools, preventing them from adapting to the planner’s intent. JADE proposes a unified architecture where all agent roles share parameters through a single LLM backbone, enabling coordinated adaptation between planning and execution. The system is trained jointly with PPO using a hybrid reward that combines global shared rewards with local penalties. Experiments on seven open-domain QA benchmarks show improvements over prior methods, with reported state-of-the-art performance.

**Compliance With Llm Reviewing Policy:**

Affirmed.

**Final Justification:**

Thanks for the rebuttal. My concerns are addressed. I will keep my original positive score and have updated my confidence to 4.

**Key Questions For Authors:**

1. Can you provide a more formal analysis of why parameter sharing specifically addresses the strategic-operational mismatch? What guarantees exist for improved coordination?

**Limitations:**

Yes

**Strengths And Weaknesses:**

**Strength**:

Soundness:
1. The approach is technically well grounded. The shared-parameter MSMDP formulation and joint optimization using PPO with a unified experience buffer are clearly defined.
2. The reward design that combines global cooperation rewards with role-specific penalties provides a reasonable mechanism for balancing overall system performance and individual agent responsibilities.
3. The empirical evaluation covers multiple benchmarks with competitive baselines and statistical reporting, which supports the paper's main claims.

Presentation:
1. The paper is clearly written and logically organized. The motivation for addressing the strategic and operational mismatch is well explained.
2. The narrative progresses smoothly from problem formulation to methodology and experiments.
3. Figures and consistent notation help communicate the system architecture and workflow.

Significance:
1. Addressed an important challenge in agentic RAG systems, namely the coordination between planning and execution.
2. The unified framework that enables interaction between different agent roles may improve efficiency and adaptability in practical systems.
3. The consistent improvements across several QA benchmarks suggest potential for broader applicability.

Originality:
1. The shared-parameter multi-agent design and the joint optimization of planning and execution introduce a novel perspective for agentic systems. The idea that role specialization can arise through prompting rather than separate models is an interesting contribution.

**Weakness**:

Technical Considerations:
1. The paper could benefit from additional discussion of the training dynamics and stability when multiple agent roles share parameters under PPO.
2. The potential trade-off between parameter sharing and role specialization could be explored further in ablation (computational cost analysis: No wall-clock time or FLOPs comparison)

Experimental Scope:
1. While the baseline comparisons are strong, including more recent agentic RAG approaches could further strengthen the evaluation.
2. Most experiments focus on open-domain factual QA, so it would be interesting to see how the framework performs on tasks (such as GAIA, Xbench) requiring longer reasoning chains. Would the RL training still be stable under such agentic information-seeking setting?
3. The choice of PPO as the optimization algorithm is not well justified. The paper would benefit from experiments comparing other RL algorithms

---

> ### Author Rebuttal · Authors · 2026-03-30
>
> # To TC1
> We would like to clarify that **shared-parameter training is not an unusual design choice**: it is widely used in cooperative MARL (e.g., QMIX, MAPPO), and is especially natural for LLM-based agents, since LLMs are inherently multi-task and role-switchable. In fact, monolithic agentic-search methods such as Search-R1 also rely on a single model to handle all roles. **We also note that the rationale for parameter sharing is discussed in more detail in Appendix A.**
>
> In JADE, training is more stable because we further **decompose** the complex agentic search process into simpler role-specific behaviors, while **jointly training all roles** instead of only optimizing the planner. This makes each agent’s function easier, provides more grounded reward signals, and avoids the planner–executor mismatch caused by decoupled optimization.
>
> We also verified this empirically through 10 independent runs (1000 steps each). JADE showed 0/10 collapses at 200, 500, and 1000 steps, while Search-R1 collapsed in 1/10, 7/10, and 9/10 runs, respectively. This provides direct evidence that shared-parameter PPO training in JADE is highly stable in practice.
> | Method | @200 Steps | @500 Steps | @1000 Steps |
> |-|-:|-:|-:|
> | Search-R1 | 1/10 (10%) | 7/10 (70%) | 9/10 (90%) |
> | **JADE (Ours)** | **0/10 (0%)** | **0/10 (0%)** | **0/10 (0%)** |
>
> # To TC2
> Thank you for this insightful comment. Without parameter sharing, there are two main alternatives: (1) assigning each agent its own full LLM, or (2) sharing the backbone but using separate LoRA adapters for different agents.
>
> The first is highly inefficient in our setting: JADE has 6 LLM-based agents, so it would require deploying 6 separate LLMs for one search task, causing substantial resource overhead and low GPU utilization.
>
> For the second, we performed an ablation with separate LoRA adapters. The results are very close to our fully shared setting:
>
> | Dataset | Full-Parameter + Shared (Ours) | Separate LoRA Adapters |
> |-|-:|-:|
> | NQ | 59.45 | 59.28 |
> | HotpotQA | 57.02 | 57.19 |
>
> **This ablation supports that parameter sharing is a reasonable design choice: it achieves essentially the same performance as separate LoRA adapters, while being simpler and more efficient in deployment.**
>
> # To ES1
> In addition to the baselines already included in Table 2, we further compared JADE with several more recent agentic search / agentic RAG baselines, including Search-o1, R1-Searcher, StepSearch, ReasonRAG, and TIPS. As shown below, JADE consistently outperforms these recent methods across all five shared benchmarks, which further strengthens the empirical support for the effectiveness of our approach.
>
> | Method | NQ | HotpotQA | 2Wiki | Bamboogle | Musique |
> |-|-:|-:|-:|-:|-:|
> | Search-o1 (EMNLP 25) | 32.4 | 44.9 | 39.7 | 37.6 | 20.1 |
> | R1-Searcher | 41.2 | 55.6 | 50.6 | 54.6 | 27.7 |
> | StepSearch (EMNLP 25) | 39.8 | 50.0 | 44.4 | 50.3 | 27.5 |
> | ReasonRAG (NeurIPS 25) | 39.7 | 41.5 | 36.6 | 40.5 | 16.4 |
> | TIPS (ICLR 26) | 40.4 | 46.0 | 43.0 | 46.2 | 18.6 |
> | **JADE (Ours)** | **59.5** | **57.0** | **53.9** | **58.3** | **29.3** |
>
> # To ES2
> Thank you for this important suggestion. **JADE is a multi-agent RL optimization framework, and therefore requires suitable training data for stable optimization.** For long-horizon agentic information-seeking benchmarks such as GAIA or Xbench, the public setups are primarily evaluation-oriented and typically do not provide corresponding training data. As a result, evaluating JADE on such tasks would either require a training-free setting, or an additional stage of synthesizing high-quality long-horizon training trajectories before MARL optimization. **We believe such long-horizon data construction is itself a substantial research problem, and is beyond what can be completed reliably within the rebuttal period.** **Nevertheless, we agree this is an important direction, and we will highlight long-horizon agentic search as a key avenue for future work.**
>
> # To ES3
> Thank you for this important comment. Our setting is a **multi-agent joint workflow**, where each query involves multiple sequential input/output interactions across different agent roles. PPO is therefore a natural choice, since it can optimize long-horizon trajectories with credit assignment over the full workflow. As shown in Figure 2, PPO also naturally supports a unified experience buffer, which is well suited to joint optimization across multiple agents.
>
> In contrast, group-based methods such as GRPO are more suitable for single-turn input/output tasks. In our setting, applying group rollouts at each step of a multi-agent workflow would cause the rollout cost to grow rapidly along the trajectory, making training much less practical. Therefore, PPO is a more suitable choice for our setting, and we will clarify this motivation in the revision.
>
> We will clarify this motivation more explicitly in the revision.
>
> **We hope our response addresses your concern.**

---

> > ### Author Rebuttal · Reviewer_dp4Y · 2026-04-01
> >
> > Thanks for the rebuttal. My concerns are addressed. I will keep my original positive scores and have updated my confidence to 4.

---

### Decision · Program_Chairs · 2026-04-30

**Decision:**

Accept (regular)

**Comment:**

The paper proposes the JADE framework, which addresses an issue of mismatch between a flexible planner and a fixed executor in the dynamic multi-agent systems. JADE uses a unified framework to enable coordinated adaptation between the planner and the executor. All four reviewers are positive towards the paper, believing that the paper has solid contribution. The authors' rebuttal further addressed several weakness pointed out by the reviewers, and all reviewers are generally satisfied with the authors' replies. The paper meet the bar to be resented at ICML, and I hope that the authors can provide a thorough revision addressing all concerns from the reviewers.